# Biochars and Activated Biocarbons Prepared via Conventional Pyrolysis and Chemical or Physical Activation of Mugwort Herb as Potential Adsorbents and Renewable Fuels

**DOI:** 10.3390/molecules27238597

**Published:** 2022-12-06

**Authors:** Małgorzata Wiśniewska, Kacper Rejer, Robert Pietrzak, Piotr Nowicki

**Affiliations:** 1Department of Radiochemistry and Environmental Chemistry, Faculty of Chemistry, Institute of Chemical Sciences, Maria Curie-Sklodowska University in Lublin, M. Curie-Sklodowska Sq. 3, 20-031 Lublin, Poland; 2Laboratory of Applied Chemistry, Faculty of Chemistry, Adam Mickiewicz University in Poznań, Uniwersytetu Poznańskiego 8, 61-614 Poznań, Poland

**Keywords:** mugwort, biochar, activated biocarbons, pyrolysis, activation, adsorption

## Abstract

The main objective of this study was to prepare a series of biochars and activated biocarbons via conventional pyrolysis as well as chemical or physical activation of solid residue after solvent extraction of wild growing plant (popular weed)–mugwort. The influence of the variant of the thermochemical treatment of the precursor on such parameters as elemental composition, textural parameters, acidic-basic character of the surface as well as adsorption abilities of the prepared carbonaceous materials was checked. Moreover, the suitability of the biochars prepared as renewable fuels was also investigated. It has been shown that the products obtained from the mugwort stems differ in many respects from the analogous materials obtained from mugwort leaves. The products were micro/mesoporous materials with surface area reaching 974.4 m^2^/g and total pore volume–1.190 cm^3^/g. Surface characterization showed that chemical activation with H_3_PO_4_ results in the acidic character of the adsorbents surface, whereas products of pyrolysis and especially physical activation show strongly alkaline surface properties. All the adsorbents were used for methylene blue and iodine adsorption from the aquatic environment. To understand the nature of the sorption process, the Langmuir, Freundlich and Temkin isotherm models were employed. The Langmuir model best described the experimental results, and the maximum sorption capacity calculated for this model reached 164.14 mg of methylene blue per gram of adsorbent. In case of iodine removal, the maximum capacity reached 948.00 mg/g. The research carried out for the biochars prepared via conventional pyrolysis showed that the value of their heat of combustion varies in the range from 21.74 to 30.27 MJ/kg, so they can be applied as the renewable fuels.

## 1. Introduction

Research on the manufacturing, physicochemical properties and possibilities of practical application of biochars and activated biocarbons have been conducted for many years by scientists from around the world. Biochars can be produced by pyrolysis, torrefaction or hydrothermal carbonization of various types of organic materials, using different time and temperature variants of thermochemical treatment [1,2,3,4,5]. In turn, for the production of activated carbons, the method of physical activation with CO_2_ and water vapor as well as chemical activation with KOH, K_2_CO_3_, ZnCl_2_ or H_3_PO_4_ are mainly used [6,7,8,9,10]. These materials owe their popularity to a well-developed specific surface, high surface activity and relatively good thermal resistance. These features have ensured that carbonaceous materials have been used for years and that new applications are still being found [11,12,13,14,15,16,17]. The number of possible precursors that can be used to produce this type of carbonaceous adsorbents is also increasing [18,19,20,21,22,23]. The rapid development of industry and ongoing urbanization, the increasing amount of chemicals and exhausted gases emitted to water, atmosphere and soil, as well as the ecological aspects that are so important nowadays, are an inspiration to look for new solutions in this field. Solutions that will be helpful in removing the enormous amounts of pollutants produced by various industrial activities, for example by their effective adsorption on new carbonaceous materials prepared from waste plant biomass. In fact, any material rich in organic carbon can be used as a precursor of carbonaceous adsorbents. However, the optimal starting material should first and foremost be readily available and cheap. Moreover, the adsorbents produced from it should be characterized by excellent sorption properties, possibility of regeneration and reapplication. A simple method of disposal of exhausted adsorbents is also very important because sooner or later they will become solid waste requiring management. 

An interesting starting material for the production of carbonaceous adsorbents seems to be various types of weeds growing on roadsides, ditches and non-cultivated areas, for example mugwort, horsetail, chickweed, creeping thistle, white goosefoot, etc. The mugwort (*Artemisia vulgaris* L.) is one of the most common weeds in Poland, belonging to the Asteraceae species. The native areas of this plant’s occurrence are mainly Europe, a large part of Asia and the areas of North Africa. In Poland, it can be found in cities, parks, on railway tracks, as well as in uninhabited areas, fields, roadsides and ditches. Mugwort is a ruderal and very expansive plant, especially in places changed as a result of human activity. Artemisia is an undemanding plant, so it can cope with drought conditions very well. Soil or air contamination is also not a major problem for her. Its shrubs reach a height of approximately 50–230 cm and are highly branched, so mugwort can be an excellent source of renewable biomass. The essential oils contained in mugwort (rich in cineole and thujone) are widely used in the cosmetic and herbal industry [24], therefore it is often subjected to a solvent extraction process, as a result of which significant amounts of solid waste are generated that must be managed.

In the presented study, we try to assess the usefulness of the mugwort herb post-extraction residue as biochars and activated biocarbons precursor and to investigate the physicochemical and sorption properties of the carbonaceous material obtained in this way. The influence of the variant of the thermochemical treatment of the precursor (conventional pyrolysis at different temperatures or chemical and physical activation) on such parameters as the elemental composition, the type of porous structure or the acidic-basic nature of the surface was investigated. The obtained carbonaceous materials were tested as potential adsorbents of methylene blue (cationic organic dye) and iodine (representing inorganic pollutants). Additionally, the usefulness of the biochars prepared as renewable fuels was assessed.

## 2. Results and Discussion

### 2.1. Physicochemical Properties of the Prepared Adsorbents

Analysis of the data presented in Figure 1 shows that the mugwort herb is characterized by a fairly high proportion of inorganic substances, however, the ash content in individual parts of the plant is very diverse. As can be seen, the proportion of mineral admixtures in mugwort stems (MS) is approximately 1 wt. % lower than in leaves and inflorescences (ML). The ash content in the structure of carbonaceous materials increases significantly as a result of the thermochemical treatment applied during the pyrolysis and the activation of both precursors. The highest mineral admixtures contribution (exceeding 22 wt. %) was found in the MLDA activated biocarbon obtained via direct physical activation of the residue after solvent extraction of mugwort leaves. In turn, the products of chemical activation (MLCA and MSCA) are characterized by an approximately two times lower ash content than the MLDA sample. In case of the biochars prepared, the ash content is determined both by the type of precursor used and the pyrolysis process temperature. The materials obtained at 300 °C (ML300 and MS300) are characterized by a significantly lower contribution of the mineral substance in the structure than the analogous samples ML400 and MS400 subjected to pyrolysis at the higher temperature (400 °C). Moreover, regardless of the thermal conditions of the conventional pyrolysis stage, biochars obtained from the mugwort leaves contain considerably more ash, than materials obtained from mugwort stems. 

On the basis of the data presented in Table 1, it can be concluded that residue after solvent extraction of leaves and stems of the mugwort differs quite significantly in terms of the content of elemental carbon, hydrogen and individual heteroatoms. The applied thermochemical transformations significantly interfere with the percentage of individual elements, radically increasing or decreasing their content (by several–over a dozen % by weight) in relation to the results obtained for the starting materials.

The greatest content of elemental carbon and, at the same time, the lowest contribution of oxygen in the structure of all tested materials are characteristic for MSCA and MSDA samples, i.e., activated biocarbons obtained by chemical and physical activation of the residue after extraction of mugwort stems. On the other hand, the analogues biocarbons prepared via activation of mugwort leaves are characterized by a higher content of nitrogen, hydrogen and oxygen in the structure. Very similar dependencies were observed for materials prepared by conventional pyrolysis of both post-extraction residues. The obtained biochars are characterized by a much higher content of elemental carbon than the corresponding precursors, but it does not exceed 80% by weight. The thermal treatment of the post-extraction residues carried out at 300 °C, and especially at 400 °C caused a significant decrease in the contribution of oxygen and hydrogen in the structure. It is a consequence of removal of volatiles present in the precursor structure and partial aromatization of the carbonaceous structure. On the other hand, the content of N^daf^ and S^daf^ in the pyrolysis products is higher than in the starting materials.

Very important parameters of carbonaceous materials (especially from the adsorption point of view) are the size of the specific surface area, the total pore volume and the type of the porous structure generated. The data summarized in Table 2 show that, independently of the type of precursor used and the thermal conditions of the pyrolysis process, the obtained biochars have a small specific surface area (7.0–16.2 m^2^/g) and a poorly developed porous structure dominated by mesopores (the average pore size is approximately 8 nm). The least favorable in this respect is the sample ML300 obtained from the residue after extraction of mugwort leaves, which was subjected to thermal treatment in an inert gas atmosphere at a temperature of 300 °C. The increase of the pyrolysis temperature by 100 °C (especially in case of mugwort leaves) resulted in an improvement in the textural parameters of biochars, but the obtained results are still not very favorable. For this reason, the paper does not include diagrams presenting nitrogen sorption isotherms and pore size distribution for these carbonaceous materials.

The further analysis of the obtained data shows that the activation of post-extraction residues turned out to be much more effective in terms of the porous structure development. The MSCA sample, obtained by chemical activation of mugwort stems, has the greatest specific surface area (974.4 m^2^/g) and, at the same time, the greatest total pore volume (1.190 cm^3^/g) among all tested materials. Interestingly, the analogous MLCA biocarbon (prepared from the mugwort leaves) has almost a three times smaller specific surface area, which indicates significant differences in the structure of individual parts of mugwort herb and their susceptibility to the development of porosity during activation stage. It can also be noticed that direct physical activation (despite the much higher processing temperature) has a much less positive effect on the surface area development than activation with H_3_PO_4_. In case of the samples activated with carbon dioxide (MLDA and MSDA), the previously observed relationship is repeated—the sample obtained from the mugwort stems has a much better developed surface area and porous system, and thus it should show better sorption ability towards potential pollutants. 

The data presented in Figure 2 and Figure 3 show that the activation products differ also in the type of porous structure generated. As can be seen, the isotherms obtained during low-temperature nitrogen sorption for MLDA and MSDA samples (Figure 2) have a very similar shape, which is close to the isotherm type I according to the IUPAC classification, which means that these materials have a large number of micropores and small mesopores in their structure.

This is confirmed, inter alia, by the micropore contribution in the total pore volume and the mean pore size values presented in Table 2 (especially in case of the sample obtained from the residue after extracting essential oil from mugwort stems). A completely different shape of isotherms was observed for both products of chemical activation. In the case of MLCA and MSCA samples, their course is close to the type IV isotherm. There is a characteristic broad hysteresis loop here, which indicates the presence of a significant amount of mesopores in the structure of the produced activated biocarbons. This finding is confirmed both by the data summarized in Table 2 as well as the by the course of the pore size distribution curves for the obtained carbon materials, presented in Figure 3. These data clearly show that the products of chemical activation have a much greater volume of pores with diameter within the range of 2–20 nm than the corresponding biocarbons obtained during activation with carbon dioxide.

Based on the analysis of the data presented in Table 3, it can be concluded that each type of thermochemical treatment of the post-extraction residues, i.e., pyrolysis or activation, reduced the content of surface acidic functional groups. This is especially evident for physically activated samples that do not contain any acidic functional groups on their surface. A slightly different trend of changes was noted in the case of the groups of a basic nature. The samples subjected to chemical activation with ortho-phosphoric acid are characterized by the lower content of this type of functional species, than the corresponding precursors. On the other hand, pyrolysis and direct physical activation contributed to a significant increase in the amounts of basic functional groups on the surface of the prepared carbon materials. The greatest increase (by 3.78 mmol/g) was noticed for the MLDA sample obtained as a result of physical activation of the residue left after mugwort leaves extraction. This may be related to the very high ash content in the structure of this activated biocarbon (above 22 wt. %, Figure 1), which is known to be alkaline in nature.

This is confirmed by the very high pH value of the water extract of this material, equal to 12.47. For the analogous MSDA activated biocarbon obtained from the mugwort stems as well as for all the biochars obtained by pyrolysis (characterized by a much lower contribution of mineral admixtures in the structure), the pH value varies in the range 10.12–8.46. On the other hand, in case of materials obtained via chemical activation, the pH value of the aqueous extracts is significantly lower and ranges from 3.23 to 3.61, which may indicate the presence of strong acidic functional groups on their surface.

### 2.2. Adsorption Properties of the Prepared Biochars and Activated Biocarbons

The information on the applicability of the carbonaceous adsorbents for removal of pollutants of molecular size close to 1 nm can be obtained from the so called iodine number or iodine value (determined by iodine adsorption from aqueous solution). According to the results presented in Figure 4, the highest iodine value (948 mg/g_ads_) was obtained for the MSCA sample, characterized by the most developed specific surface area and porous structure among the produced materials. This result exceeds the iodine number for commercial activated carbons, such as WG-12 (produced from hard coal by Gryfskand, Poland), SX2 (produced from peat by Norit Activated Carbon, The Netherlands) or Filtrasorb 300 (produced from bituminous coal by Calgon Carbon Corporation, Pittsburgh, PA, USA), which are commonly used in the decolorization and/or treatment of drinking water. Unfortunately, in case of others activated biocarbons (especially MLDA and MSDA samples) the results are much less satisfactory (iodine adsorption below 500 mg/g). Noteworthy are also high values of iodine numbers obtained for biochars, in particular for ML300 (671 mg/g) and ML400 (774 mg/g) samples, prepared via pyrolysis of the residue of mugwort leaves after solvent extraction in both temperature variants. Their sorption capacity towards iodine turned out to be better than for MLDA and MSDA samples, obtained by the direct physical activation method, consequently they can be used as potential bio-sorbents.

Another method of determining the adsorption properties of the prepared carbonaceous materials was the assessment of the ability to remove organic impurities from aqueous solutions, based on the adsorption of a synthetic cationic dye—methylene blue. The results of the adsorption tests are presented in Table 4 and Table 5 as well as in Figure 5.

The products of physical activation turned out to be the least effective adsorbents against methylene blue, especially the MLDA sample, which was able to adsorb less than 14 mg of this organic compound from an aqueous solution. What is more, the efficiency of methylene blue removal for this adsorbent is only 67% even at the initial dye concentration equal to 5 mg/cm^3^. Although the sorption capacity of the analogous MSDA activated biocarbon (obtained from mugwort stems post-extraction residue) is almost twice as high, it is still not a satisfactory result in terms of possible practical application. The obtained biochars, in particular ML300 and ML400 samples, are much more effective in terms of organic dye adsorption. Both of the mentioned materials show approximately a six times higher sorption capacity in relation to methylene blue (84.09 and 85.74 mg/g, respectively) than the activated biocarbon obtained by physical activation of the same precursor. It is quite interesting that in case of analogous carbon materials prepared from mugwort stems, no similar relationship was found. The sorption capacity of MS300 and MS400 biochars is just approximately 1 mg higher than for MSDA activated biocarbon.

The situation is completely different for materials obtained via chemical activation with H_3_PO_4_. In this case, post-extraction residue of mugwort stems turned out to be a better precursor, and the MSCA sample obtained in this way is able to adsorb almost 164.14 mg of methylene blue, which allows it to compete with commercial products, such as Norit SX2 (161 mg/g) or WG-12 (190 mg/g), obtained from peat and hard coal, respectively. According to the data presented in Figure 5, the MSCA sample shows 100% efficiency in methylene blue removal in a quite wide range of its initial concentrations, i.e., ranging from 5 to 60 mg/dm^3^.

In Table 4 and Table 5, apart from the maximum sorption capacity, the constants characteristic for the adsorption models according to Langmuir, Freundlich and Temkin are presented. As it is commonly known, the applied model describes the course of the pollutant adsorption process in more detail if the value of the correlation coefficient R^2^ is higher, i.e., closer to one. The obtained data show that for the majority of the tested carbonaceous adsorbents, the highest values of the correlation coefficient were obtained for the Langmuir model, which may indicate that adsorption of methylene blue proceeds with the formation of the so-called adsorbate monolayer on the surface of the carbon materials and adsorption of each dye molecule has equal activation energy. Nevertheless, for several of the tested adsorbents (e.g., MSDA, ML400 or MS 400) equally high R^2^ values were obtained for the Freundlich model, which may suggest that the methylene blue adsorption mechanism is much more complex and its precise determination requires further research. Only in case of the MLDA sample, the experimental data were the best described by the Freundlich model (assuming multilayer adsorption process on heterogeneous adsorption sites), which was indicated by much higher value of the correlation coefficient than for the Langmuir and Temkin model. In turn, in case of the ML300 biochar, the highest value of the determination coefficient (R^2^ = 0.987) was found for the Temkin model, which is assuming that the heat of adsorption of all the molecules in the layer decreases linearly with coverage due to adsorbent–adsorbate interactions. However, this model is ignoring extremely low and high analytes concentrations.

Based on the analysis of the data presented in Table 6, it can be concluded that the MS400 and MSCA samples perform well in terms of iodine and methylene blue adsorption compared to carbonaceous materials obtained from other types of biomass.

The sorption capacity of both materials towards iodine is significantly lower only in comparison with activated carbon prepared by chemical activation of bean husk with H_3_PO_4_ at the weight ratio 1:5. Evwierhoma et al. have proved that this material is able to adsorb 1256 mg of iodine per g of adsorbent [25]. However, it should be emphasized that biochar obtained by pyrolysis of mugwort leaves post-extraction residue at 400 °C shows a significantly higher efficiency of iodine adsorption from an aqueous solution than activated carbons obtained from Spinacia oleracea (spinach) leaves [26], coconut shells [27], tamarind seeds [28] or Acacia wood [29]. Of course, the MSCA activated biocarbon is even more favorable in this comparison. Adsorbents obtained from mugwort post-extraction residue also perform quite well when we compare their methylene blue sorption capacity with other carbonaceous materials. Unfortunately, they do not achieve such spectacular results as the biochar derived from soybean dreg by Ying et al. [31] via one-pot synthesis, the capacity of which towards the aforementioned dye is as much as 1274 mg/g. The materials obtained (especially biochar ML400) also perform worse than the activated carbon prepared from Citrullus lanatus rinds (262 mg/g). However, MSCA activated carbon, obtained via chemical activation of mugwort post-extraction residue, is able to adsorb much more methylene blue (167 mg/g) than materials obtained from tamarind seed [28], *Gigantochloa* bamboo [32], cashew nut shells [33] and lignocellulosic agriculture wastes [34], whose sorption capacity ranges from 87 mg/g to 149 mg/g.

### 2.3. Energy Parameters of the Biochars Prepared

Analysis of the biochars energy parameters allowed to determine whether carbonaceous materials obtained as a result of conventional pyrolysis of mugwort post-extraction residues can be applied as renewable fuels. The data presented in Figure 6 and Figure 7 shows that the mass yield of the product (MY), energy densification ratio (EDR), energy yield (EY) as well as higher heating values (HHV) for the obtained biochars are to a large extent determined by the pyrolysis temperature as well as the type of starting material used for their production. Both for the carbonaceous materials prepared from the mugwort leaves and stems, the higher mass yield of the pyrolysis process was obtained at a temperature of 300 °C. However, the stems turned out to be slightly more resistant to high temperature treatment, as the yield of the final pyrolysis product was approximately 4 wt. % higher than for the analogous biochars obtained from the mugwort leaves. For the other energetic parameters, an opposite relationship was observed, i.e., higher values of EY, EDR and HHV were obtained in case of the biochars subjected to conventional pyrolysis at 400 °C. 

The highest value of the heat of combustion (30.27 MJ/kg), was shown by the MS400 sample obtained via pyrolysis of the mugwort stems post-extraction residue at a temperature of 400 °C. It should be emphasized that this result is comparable to the HHV values obtained for a good quality hard coals (31–29 MJ/kg) and higher than for typical lignite briquettes, wood and wood pellets (~25–18 MJ/kg). However, it should be noted that commercial raw materials are not subjected to high-temperature pre-treatment. In case of the other biochars, the value of the heat of combustion was much lower and ranged from 25.29 MJ/kg to 21.74 MJ/kg. On the basis of the obtained data, it can also be concluded that regardless of the temperature applied during the pyrolysis process, the biochars obtained from the mugwort stems show more favorable energy parameters than analogous materials prepared from mugwort leaves post-extraction residue. This is especially visible when comparing the HHV values for MS400 and ML400 samples, where the difference is almost 5 MJ/kg.

According to the data presented in Table 7, the best of the obtained biochars (MS400 sample) shows favorable energy parameters not only in comparison to commercial products but also to carbon materials prepared from a various kinds of biomass. It is clearly inferior only to biochars prepared from groundnut shells [36], coconut and palm shells [37], as well as wood chips [38], for which the HHV value is 45.2, 33.7, 33.6 and 33.2 MJ/kg, respectively.

## 3. Materials and Methods

### 3.1. Biochars and Activated Carbons Preparation

The precursor of the biochars and activated biocarbons was solid residue generated in the solvent extraction of mugwort inflorescences and leaves (ML, Figure 8a) or mugwort stems (MS, Figure 8b). The mugwort herb was harvested in 2021 in the Wielkopolska region (the western part of Poland) and next air-dried and crushed.

In the first step of biochars preparation, both precursors were dried at 110 °C to a constant mass and then subjected to conventional pyrolysis in a nitrogen atmosphere (technical nitrogen 4.0, Linde Gaz Poland, Kraków, Poland) using horizontal laboratory furnace equipped with a quartz tubular reactor (one-heating-zone model PRW75/LM, provided by Czylok, Jastrzębie-Zdrój, Poland). Approximately 20 g of the starting materials were placed in the nickel boats, heated to 300 or 400 °C (at the rate of 5 °C/min) and annealed at the final pyrolysis temperature for 30 min. After the pyrolysis stage the samples were cooled down in nitrogen atmosphere with the flow rate of 170 cm^3^/min. The obtained biochars (Figure 9) were denoted as ML300, ML400, MS300 and MS400, respectively.

Another part of the post-extraction residue was chemically activated (CA) with phosphoric(V) acid. For this purpose, the starting materials were impregnated with 50% H_3_PO_4_ solution (POCH, Gliwice, Poland) at the precursor: activating agent weight ratio of 1:2. After 24 h of the impregnation stage (at room temperature), the samples were dried at 120 °C to complete evaporation of water. Next the impregnated precursors were placed into the quartz boats and heated under nitrogen atmosphere (flow rate 250 cm^3^/min). In the first step, the samples were heated to the temperature of 200 °C (at the rate of 5 °C/min) and annealed at that temperature for 30 min. In the next stage, the samples were heated to the final activation temperature of 550 °C (also at the rate of 5 °C/min) and again thermostated for 30 min. After that, the products of chemical activation were cooled down to room temperature in nitrogen flow. The activated carbons prepared in such a way were washed with 20 dm^3^ of boiling distilled water on a vacuum filtration funnel with a built-in sintered glass disc (in order to remove excess of the activating factor and side products) and finally dried to constant mass at 110 °C. The obtained activated carbons were denoted as MLCA and MSCA.

The remaining part of the plant material was placed in the nickel boats and then subjected to direct activation (DA), i.e., single-stage physical activation (excluding the pyrolysis stage). The boats were placed in a laboratory furnace preheated to a temperature of 800 °C for a period of 45 min. Carbon dioxide (technical CO_2_ 2.8, Linde Gaz Polska) was used as the activating agent, which flowed through the tubular quartz reactor at the rate of 250 cm^3^/min. After the designated activation time had elapsed, the boats were removed from the hot zone of the furnace and cooled down to room temperature under CO_2_ flow. The activated samples were next washed with boiling distilled water and dried to constant weight at 110 °C. The obtained activated carbons were denoted as MLDA and MSDA.

### 3.2. Characterization of the Biochars and Activated Carbons

The elemental analysis of the starting post-extraction residue, biochars as well as activated biocarbons prepared was performed using the Vario EL III elemental analyzer provided by Elementar Analysensysteme GmbH (Langenselbold, Germany). This analysis consisted of the catalytic combustion of the carbonaceous materials at a temperature of 1200 °C and a detection of exhausted gases based on the difference in thermal conductivity. The total ash content for all the materials under investigation was determined by burning the samples (in a form of powder with particle size below 1 mm) in a microwave muffle furnace (model Phoenix, CEM Corporation, Matthews, IL, USA) at 815 °C for 1 h (PN-ISO 1171:2002 Standard). For each sample, two parallel combustions were carried out.

The porous structure of biochars and activated biocarbons was characterized on the basis of low-temperature N_2_ adsorption/desorption isotherms measured at –196 °C on the sorptometer Autosorb iQ, provided by Quantachrome Instruments (Boynton Beach, FL, USA). Prior to the analysis, the carbonaceous materials were degassed under vacuum at a temperature of 250 °C (in case of biochars) or 300 °C (in case of activated biocarbons) for 12 h, in order to remove all pre-adsorbed gaseous species. The specific surface area of the carbon materials was designated on the basis of the multilayer adsorption BET theory. Pore size distribution as well as total pore volume for each biochar and activated biocarbon sample were determined based on the BJH model. Additionally, the commonly known t-plot method was applied to determine micropore volume and surface area as well as external surface area.

In order to determine the content of surface functional groups of basic or acidic nature, the Boehm titration method was applied [45]. Volumetric standards of 0.1 mol/dm^3^ NaOH (POCH, Gliwice, Poland) and 0.1 mol/dm^3^ HCl (POCH, Gliwice, Poland) were used as the titrants, whereas 1% water solution of methyl orange (POCH, Gliwice, Poland) was used as an indicator. For each of the samples, 2 parallel determinations were made.

To specify the chemical nature of the surface of the tested carbonaceous materials, the pH of their water extracts was also determined, using following procedure: a portions of 1 g of the biochars or activated biocarbons were mixed with 100 cm^3^ of distilled water and stirred magnetically overnight to reach equilibrium state. After that, the pH of the suspension was measured (upon continuous stirring), using CP–401 pH-meter (ELMETRON, Zabrze, Poland), equipped in combined glass electrode EPS-1. For each of the samples, two parallel measurements were made.

### 3.3. Adsorption of Inorganic and Organic Pollutants

The iodine sorption ability of the carbonaceous adsorbents was determined according to the procedure described in PN-83/C-97555.04. Standard: portion of 0.2 g of the sample sieved to a particle size below 0.09 mm was placed in 250 cm^3^ flasks and mixed with 4 cm^3^ of 5% hydrochloric acid (POCH, Gliwice, Poland) as well as 20 cm^3^ of 0.1 mol/dm^3^ iodine water solution (POCH, Gliwice, Poland). The mixture was shaken for 4 min in a laboratory shaker, filtered through filter paper and washed with 50 cm^3^ of distilled water. The resulting solution was titrated with 0.1 mol/dm^3^ sodium thiosulphate (POCH, Gliwice, Poland) until the solution became colorless (1% starch water solution was used as the indicator). For each of the biochars and activated biocarbons two parallel determinations were made.

Adsorption of methylene blue (POCH, Gliwice, Poland) from aqueous solution was performed using the following procedure. Samples of the prepared biochars and activated biocarbons in the same portions of 0.025 g (particle size of 0.1 mm) were added to 50 cm^3^ of the dye solution with the initial concentrations in the range from 5 to 150 mg/dm^3^ and then the suspensions were stirred magnetically (150 rpm) for 8 h, at a temperature of 22 ± 1 °C. After the adsorption equilibrium state had been achieved, the solutions were separated from the carbonaceous materials by centrifugation at 5000 rpm, for 5 min (with a Frontiner ™ centrifuge FC5515 OHAUS, Parsippany, NJ, USA) in order to minimize the amount of carbon dust in the analyte collected for spectroscopic analysis. The equilibrium methylene blue concentration was established spectrophotometrically applying a double beam Cary 100 Bio UV–Vis spectrophotometer (provided by Agilent, Santa Clara, CA, USA) at the wavelength of 664 nm (using the previously prepared calibration curve). All experiments were made in duplicate.

The equilibrium adsorbed amount of methylene blue (*q_e_*, mg/g) was calculated according to the following Formula (1):(1)$$q_{e}=\frac{c_{ads}\cdot V_{dye\;solution}}{m_{sample}}\;$$
where *c_ads_*–the difference in the methylene blue concentration in the system before and after its adsorption [mg/dm^3^], *V_dye solution_*–the volume of the methylene blue water solution [dm^3^], *m_sample_*–is the mass of biochar or activated biocarbon used [g].

The effectiveness of removal (*ER*) of methylene blue from its water solutions was calculated from the following Formula (2):(2)$$ER=\frac{c_{init}-c_{eq}}{c_{init}}\cdot 100\%\;$$
where *c_init_*–the initial methylene blue concentration [mg/dm^3^], *c_eq_*–the equilibrium methylene blue concentration [mg/dm^3^].

### 3.4. Energy Parameters of the Biochars Prepared

The higher heating value of biochars (HHV, also known as gross calorific value) was determined in accordance with the procedure presented in the ISO 1928:2009 Standard, using the KL-12Mn calorimeter, provided by PRECYZJA-BIT (Bydgoszcz, Poland). Before the measurements, all biochar samples were dried for 12 h at 110 °C to complete evaporation of adsorbed water. A portion 0.5 g of each biochar sample (in the form of compressed pellet) was placed in a crucible and then the crucible was placed in the calorimetric bomb. In order to supply enough oxygen for combustion, the measurement vessel was pressurized with pure oxygen to 2.5 MPa. Next the bomb was placed in a calorimetric vessel filled with distilled water and ignited by the iron resistance wire. The biochar sample underwent combustion and the heat generated during combustion was transferred to the surrounding water and heated it. By recording the difference in the water temperature during the whole process, the system calculated the *HHV* (MJ/kg) using the following Formula (3):(3)$$HHV_{biochar}=\frac{C_{bomb}\cdot \Delta T-Q_{iw}}{m_{sample}}\;$$
where *C_bomb_*—the heat capacity of the bomb calorimeter [MJ/°C], Δ*T*—the change of water temperature [°C], *Q_iw_*—the amount of heat generated by iron wire [MJ/kg], *m_sample_*—the mass of biochar used [kg].

Based on the precursors masses and higher heating values of the biochars prepared, the mass yield of pyrolysis process (*MY*), energy densification ratio (*EDR*) as well as energy yield (*EY*) were calculated, using the following Formulas (4)–(6):(4)$$MY=\frac{m_{biochar}}{m_{precursor}}\cdot 100\%$$
(5)$$EDR=\frac{HHV_{biochar}}{HHV_{precursor}}$$
(6)EY=MY·EDR  

## 4. Conclusions

The conducted studies have shown that mugwort herb can be used as an alternative raw material for the production of biochars and activated biocarbons. The ability to survive in unfavorable environmental conditions, high availability and pro-ecological aspects make this plant the attractive precursor. The physicochemical properties of carbonaceous materials obtained from post-extraction residues are determined by both the variant of the thermochemical treatment applied as well as the part of the plant used for their production. The products obtained from mugwort stems differ in many respects from the corresponding materials obtained from leaves. The elemental composition, the degree of the specific surface area development, the type of porous structure produced as well as the acidic-basic nature of the surface of the biochars prepared as a result of conventional pyrolysis of both precursors depend to a large extent on the thermal conditions of the process. Samples subjected to pyrolysis at 400 °C are characterized by better developed surface area and total pore volume as well as by the greater content of surface functional groups of a basic nature. In case of activated biocarbons, the decisive factor is the type of activator used. Samples activated with H_3_PO_4_ have a significantly larger surface area and their porous structure consists mainly of mesopores, whereas analogous physical activation products are characterized by a significant contribution of micropore. Moreover, the chemically activated samples show the clearly acidic character of the surface, whereas for materials activated with CO_2_ it is alkaline.

As shown by the adsorption tests, the obtained carbon materials are characterized by very diverse sorption abilities towards organic and inorganic pollutants. The most advantageous in this respect is the product obtained via chemical activation of the mugwort stems with H_3_PO_4_. Its sorption capacity towards model impurities (164.14 mg/g for methylene blue and 948.00 mg/g for iodine) reaches the level comparable or even higher than for some commercial products.

Analysis of the biochars energy parameters have shown that biochar MS400 obtained via conventional pyrolysis of mugwort stems post-extraction residue at 400 °C can be successfully used as renewable fuel. The higher heating value of this biochar (equal to 30.3 MJ/kg) is at a level similar to that of good quality bituminous coals.

## Figures and Tables

**Figure 1 molecules-27-08597-f001:**
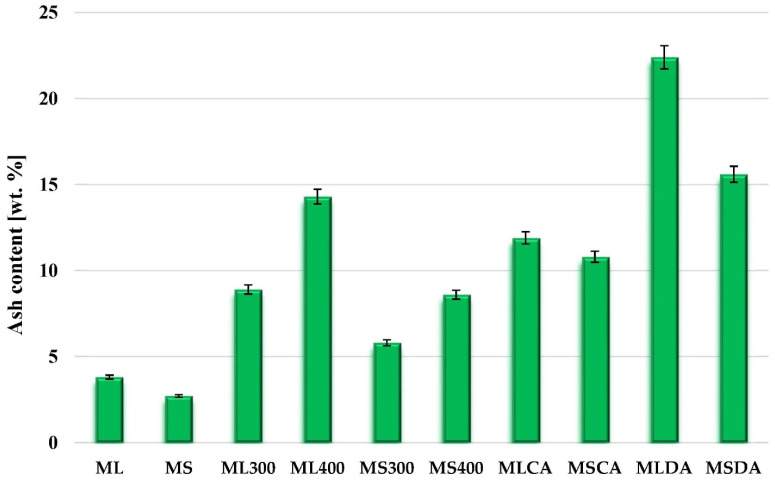
Ash content for the precursors (ML and MS), biochars (ML300, ML400, MS300 and MS400) as well as activated biocarbons obtained via chemical activation (MLCA and MSCA) or physical activation (MLDA and MSDA).

**Figure 2 molecules-27-08597-f002:**
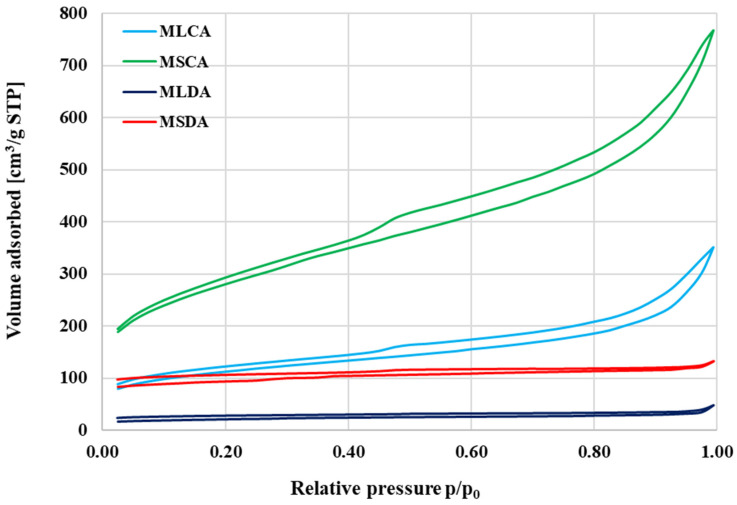
Low-temperature nitrogen adsorption/desorption isotherm for the activated biocarbons prepared from mugwort post-extraction residue.

**Figure 3 molecules-27-08597-f003:**
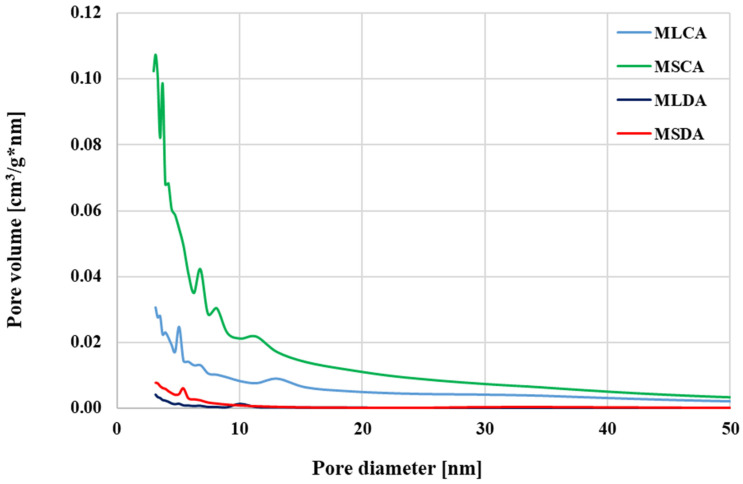
Pore size distribution for the activated biocarbons prepared from mugwort post-extraction residue.

**Figure 4 molecules-27-08597-f004:**
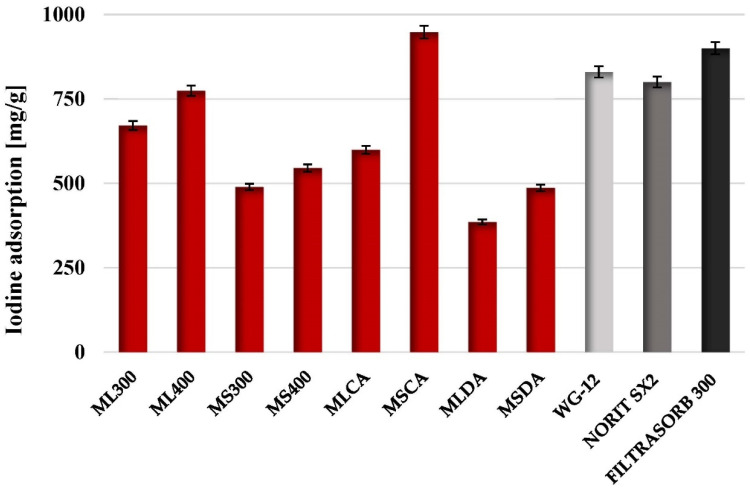
Comparison of iodine adsorption by biochars and activated biocarbons prepared from mugwort post-extraction residue with commercial products.

**Figure 5 molecules-27-08597-f005:**
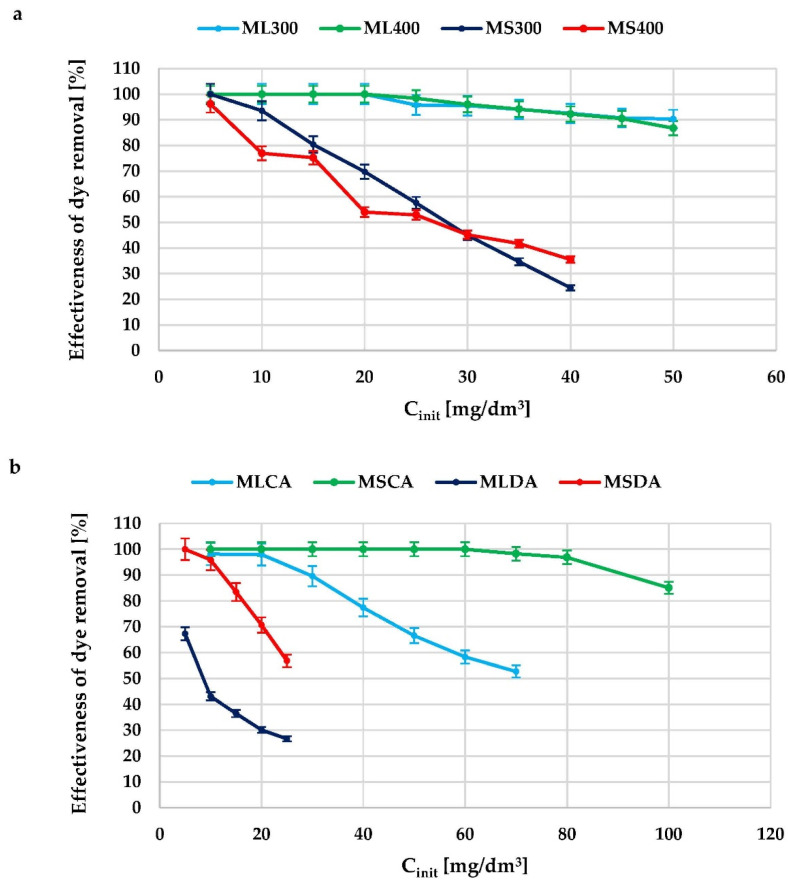
The effectiveness of methylene blue removal from aqueous solutions for the biochars (**a**) and activated biocarbons (**b**) prepared from mugwort post-extraction residue.

**Figure 6 molecules-27-08597-f006:**
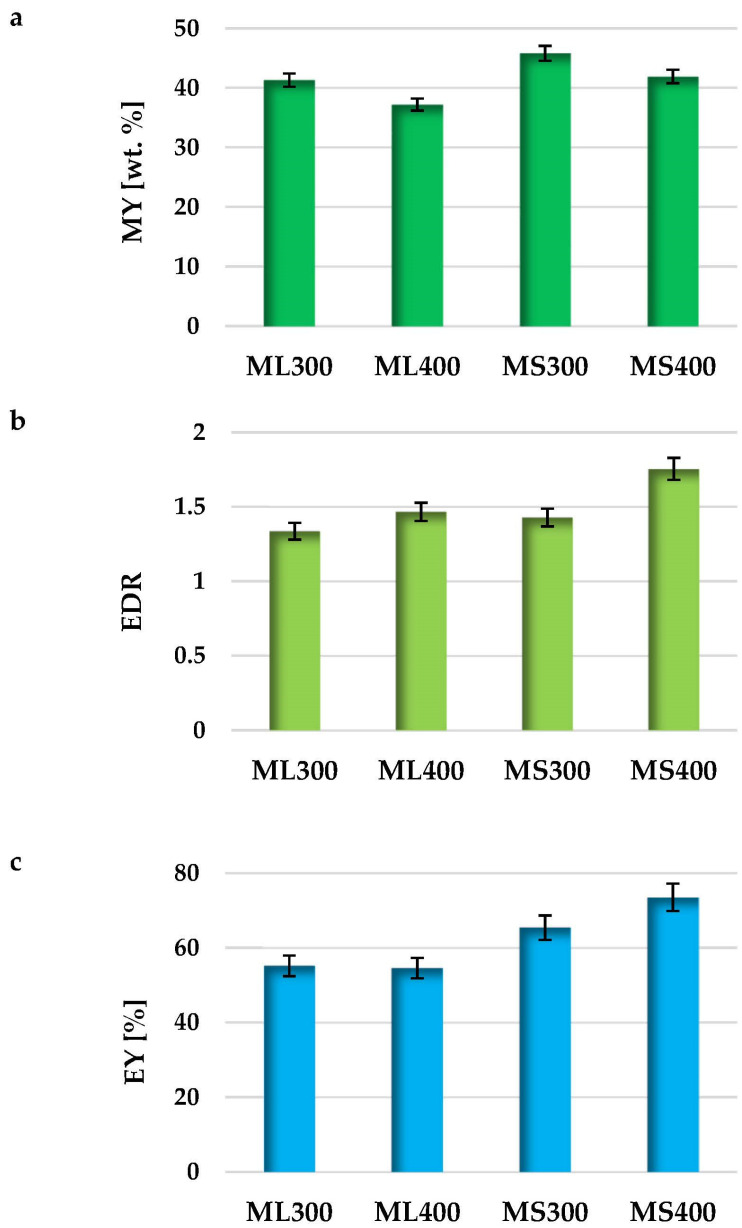
The mass yield of pyrolysis process (**a**), energy densification ratio (**b**) as well as energy yield (**c**) of the biochars prepared.

**Figure 7 molecules-27-08597-f007:**
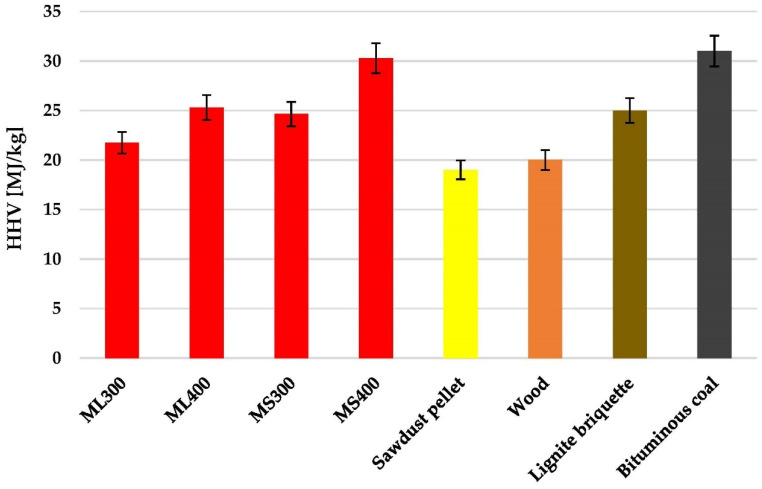
Comparison of the higher heating values of biochars prepared from mugwort post-extraction residue with commercial fuels.

**Figure 8 molecules-27-08597-f008:**
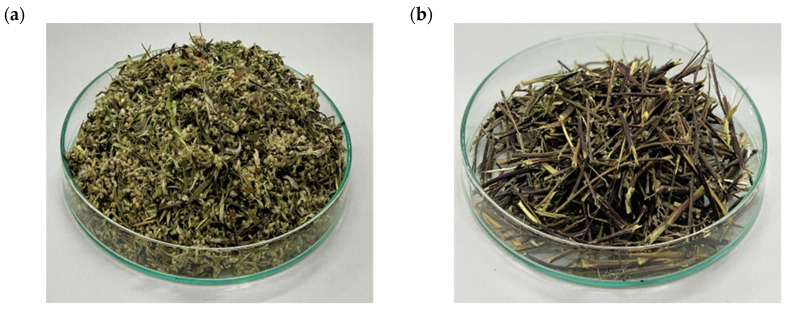
Precursor of biochars and activated biocarbons–a post-extraction residue obtained from the mugwort inflorescences/leaves (**a**) and mugwort stems (**b**).

**Figure 9 molecules-27-08597-f009:**
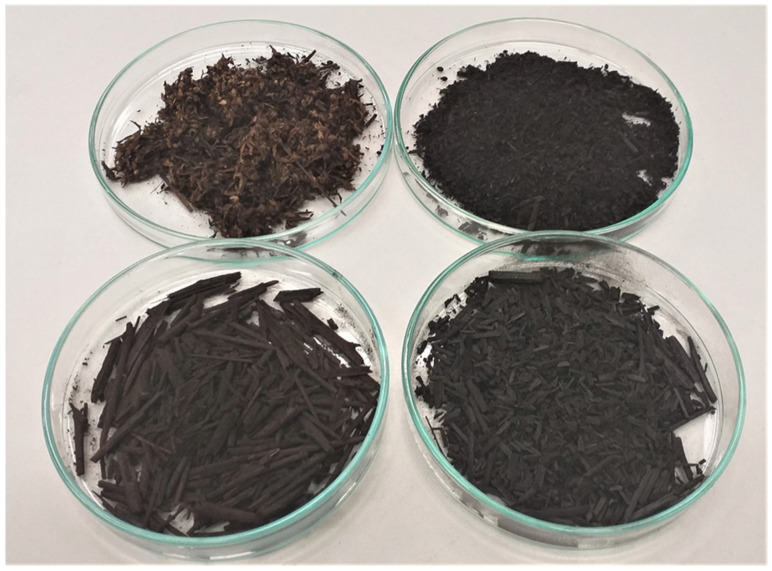
Pyrolysis products obtained at a temperature of 300 and 400 °C.

**Table 1 molecules-27-08597-t001:** Elemental composition of the precursors, biochars and activated biocarbons.

Sample	C^daf 1^	H^daf^	N^daf^	S^daf^	O^diff 2^
ML	46.0	7.8	2.5	0.3	43.4
MS	51.4	7.6	0.7	0.2	40.1
ML300	65.7	5.9	3.6	0.9	23.9
ML400	72.4	5.1	3.9	0.8	17.8
MS300	71.6	5.2	0.9	0.8	21.5
MS400	77.7	4.4	1.2	0.6	16.1
MLCA	87.5	2.6	0.9	0.8	8.2
MSCA	94.7	2.3	0.4	0.9	1.7
MLDA	82.2	0.9	3.2	1.1	12.6
MSDA	95.7	0.7	0.9	0.9	1.8

^1^ dry-ash-free basis; ^2^ calculated by difference; method error ≤ 0.3%.

**Table 2 molecules-27-08597-t002:** Textural parameters of the biochars and activated biocarbons prepared from the mugwort post-extraction residue.

Sample	Total ^1^		Micropore		Micropore Contribution	Mean Pore Size [nm]
	Surface Area [m^2^/g]	Pore Volume [cm^3^/g]	Area [m^2^/g]	Volume [cm^3^/g]		
ML300	7.0	-	0.014	-	-	8.359
ML400	16.2	-	0.032	-	-	8.115
MS300	11.6	-	0.024	-	-	8.346
MS400	15.7	-	0.031	-	-	7.795
MLCA	377.4	141.5	0.544	0.073	0.13	5.773
MSCA	974.4	204.3	1.190	0.100	0.08	4.885
MLDA	69.1	35.5	0.075	0.018	0.24	4.325
MSDA	291.4	187.8	0.205	0.101	0.49	2.822

^1^ method error in the range from 2 to 5%.

**Table 3 molecules-27-08597-t003:** Acidic-basic properties of the precursors, biochars and activated biocarbons prepared.

Sample	Acidic Groups ^1^ [mmol/g]	Basic Groups ^2^ [mmol/g]	Total Amount [mmol/g]	pH of Water Extracts
ML	1.19	0.66	1.85	6.51
MS	0.92	0.35	1.27	6.27
ML300	0.95	1.57	2.52	8.46
ML400	1.14	1.03	2.17	8.85
MS300	0.49	2.68	3.17	10.08
MS400	0.90	1.27	2.17	9.92
MLCA	1.18	0.12	1.30	3.23
MSCA	0.65	0.25	0.90	3.61
MLDA	0.00	4.58	4.58	12.47
MSDA	0.00	1.83	1.83	10.12

^1^–carboxyl, carbonyl, lactone, phenol, anhydride, ^2^–pyrone, chromene, quinones.

**Table 4 molecules-27-08597-t004:** Langmuir, Freundlich and Temkin parameters of the isotherms of methylene blue adsorption on the biochars prepared from mugwort post-extraction residue.

Isotherm Model	Parameters	Biochars			
		ML300	ML400	MS300	MS400
Langmuir	R2	0.962	0.992	0.999	0.983
	qm	84.09	85.74	29.02	28.77
	KL	0.54	0.16	0.15	0.03
Freundlich	R2	0.982	0.970	0.914	0.953
	KF	47.98	55.68	20.44	14.08
	1/n	0.388	0.240	0.131	0.233
Temkin	R2	0.987	0.758	0.625	0.919
	AT	6.17	17.82	397.26	43.84
	B	25.579	18.293	3.457	4.012

R^2^–correlation coefficients, q_max_–the maximum adsorption capacity (mg/g), K_L_–the Langmuir adsorption equilibrium constant (dm^3^/mg), K_F_–the Freundlich equilibrium constant [mg/g (mg/dm^3^)^1/n^], 1/n–the intensity of adsorption, A_T_–the Temkin equilibrium binding constant (dm^3^/mg), B–a constant equal to B = RT/B_T_; where B_T_ is the Temkin constant (J/mol), R–is gas constant (8.31 J/mol·K) and T is absolute temperature (K).

**Table 5 molecules-27-08597-t005:** Langmuir, Freundlich and Temkin parameters of the isotherms of methylene blue adsorption on the activated biocarbons prepared from mugwort post-extraction residue.

Isotherm Model	Parameters	Activated Biocarbons			
		MLCA	MSCA	MLDA	MSDA
Langmuir	R2	0.997	0.990	0.967	0.998
	qm	73.80	164.14	13.91	27.83
	KL	0.02	0.01	0.04	0.43
Freundlich	R2	0.879	0.936	0.981	0.993
	KF	37.22	102.09	5.64	22.16
	1/n	0.219	0.149	0.289	0.133
Temkin	R2	0.962	0.901	0.611	0.971
	AT	63.26	202.55	3.48	197.77
	B	10.052	19.147	3.565	4.055

**Table 6 molecules-27-08597-t006:** Adsorption capacities towards iodine and methylene blue for various carbonaceous adsorbents.

Carbonaceous Adsorbent	Maximum Adsorbed Amount [mg/g]	Reference
**Iodine**		
Activated carbon from bean husk	1256	[25]
Activated carbon obtained from Spinacia oleracea (spinach) leaves	623	[26]
Activated carbon from coconut shell	249	[27]
Activated carbon from tamarind seed	310	[28]
Activated carbon based on acacia wood	381	[29]
Corn stalk hydrothermal and pyrolytic biochars	17 and 10	[30]
Mugwort-based biochar and activated biocarbon	774 and 948	This study
**Methylene blue**		
Activated carbon from tamarind seed	96	[28]
Biochar derived from soybean dreg	1274	[31]
*Gigantochloa* bamboo-derived biochar	87	[32]
Activated carbon synthesized from cashew nut shells	100	[33]
Activated carbon derived from lignocellulosic agriculture wastes	149	[34]
Activated carbon prepared from *Citrullus lanatus* rind	232	[35]
Mugwort post-extraction residue-based biochar and activated biocarbon	84 and 167	This study

**Table 7 molecules-27-08597-t007:** Heats of combustion of various biochars.

Biochar	Heat of Combustion [MJ/kg]	Reference
Biochar produced from pyrolysis of groundnut shell	45.2	[36]
Biochars from coconut and palm shells	33.7 and 33.6	[37]
Biochar from wood chips and rice hulls	33.2 and 14.8	[38]
Biochar derived from palm pressed fibre	27.3	[39]
Biochar from waste straw	22.6	[40]
Biochars from rapeseed and poplar waste biomass	20.6 and 22.3	[41]
Biochar from almond shell	28.2	[42]
Biochars from pinewood and white straw	27.8 and 22.0	[43]
Biochars formed from miscanthus straw and sawdust	26.6 and 23.4	[44]
Mugwort post-extraction residue-based biochar	30.3	This study

## Data Availability

Data are contained within the article.

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
