# Peer review of "Biochars and Activated Biocarbons Prepared via Conventional Pyrolysis and Chemical or Physical Activation of Mugwort Herb as Potential Adsorbents and Renewable Fuels"

_molecules, 2022, doi:10.3390/molecules27238597_

Round 1

Reviewer 1 Report

This study is about "Mugwort herb as the precursor of biochars and activated bio-carbons" and ornanized well.

       The main objective of this study was to prepare a series of biochars and activated biocar-bons via conventional pyrolysis as well as chemical or physical activation of solid residue after sol-vent extraction of wild growing plant (popular weed) – mugwort. The conducted studies have shown that mugwort can be used as an alternative raw material for the production of biochars and activated biocarbons.    Results of this studies are good.  So, This study can be accepted  in present form  but considering the sentences given below

1-The main objective of this study was to prepare a series of biochars and activated biocar-bons via conventional pyrolysis as well as chemical or physical activation of solid residue after sol-vent extraction of wild growing plant (popular weed) – mugwort

2-the topic is original and relevant in the energy based on biocharg.

3-The conducted studies have shown that mugwort can be used as an alternative raw material for the production of biochars and activated biocarbons.

 4-As metodology erorbars should be added to figures for repeatibility

5. The conclusions are consistent with the evidence and arguments presented and they address the main aim of study
6. The references are appropriate
7.  Tables and figures is  suitable. But If it is avaliable, error bars should be added to figures.

Author Response

First of all, we would like to thank you for your kindly review and suggestions that allowed us to improve the manuscript.

 A metodology error bars should be added to figures for repeatibility.

According to Reviewer sugesstion error bars have been added to Figures.

Reviewer 2 Report

The format used does not have left numeration, therefore is hard to give precise feedback.

Introduction.

In line 6, please change "newer and newer"

In line 7-8, make clear the phrase "The rapid development of industry and economy"

line 11-12, rewrite the phrase "They will be helpful in the removal of enormous amounts of pollutions produced by various industrial activities".

This idea is quite messy. please rewrite it in a clearer way "A good precursor of carbonaceous adsorbents should be cheap and easily accessible material. In turn, an ideal adsorbent must first of all be material with excellent sorption properties, but the possibility of its regeneration and reapplication as well as simplicity of disposal is also very important, because with time it becomes a waste as well."

this phrase "The main goal of the study was to obtain a series of biochars and activated biocarbons by chemical and physical activation and conventional pyrolysis of post-extraction residue obtained from the stems and leaves of a wild-growing plant – mugwort." is pretty much the same as the last paragraph in the introduction section. Please be concise with this part.

Results

Figure 2 is hard to read. please explain the names in the X-axis.

Maybe the author should re-write the different acronyms. 

Table 3 is not in format, please change it

There is no discussion section

Author Response

First of all, we would like to thank you for your kindly review and importatnt suggestions that allowed us to improve our manuscript.

Introduction.

In line 6, please change "newer and newer"

It has been changed.

In line 7-8, make clear the phrase "The rapid development of industry and economy"

It has been improved.

line 11-12, rewrite the phrase "They will be helpful in the removal of enormous amounts of pollutions produced by various industrial activities".

It has been re-written.

This idea is quite messy. please rewrite it in a clearer way "A good precursor of carbonaceous adsorbents should be cheap and easily accessible material. In turn, an ideal adsorbent must first of all be material with excellent sorption properties, but the possibility of its regeneration and reapplication as well as simplicity of disposal is also very important, because with time it becomes a waste as well."

It has been changed.

this phrase "The main goal of the study was to obtain a series of biochars and activated biocarbons by chemical and physical activation and conventional pyrolysis of post-extraction residue obtained from the stems and leaves of a wild-growing plant – mugwort." is pretty much the same as the last paragraph in the introduction section. Please be concise with this part.

It has been corrected.

Results

Figure 2 is hard to read. please explain the names in the X-axis. Maybe the author should re-write the different acronyms.

According to advice of the another Reviewer Figure 2 has been changed to Table 1.

Table 3 is not in format, please change it.

It has been improved.

There is no discussion section

We would like to clarify that the discussion section is linked to the Results section. However, in the new version of the manuscript it has been expanded.

Reviewer 3 Report

Comments and suggestions

1.I propose to precisely define the title and the exact purpose of the work. The aim of the work is too general, while the title should emphasize biochars and the method of its acquisition

2. There are no conclusions in the abstract, for the reader it is not important in the foreground that the examined parameters changed in such a wide range

3. In introduction there is no reference to methods used for biochar precursor preparation which may ultimately affect the activated carbon properties (surface area, porosity, and other)

4. Please provide reference methods for the characterization of biochars and activated carbons. When setting parameters, how many repetitions were made

5. Has the Temkin model been considered which considers the interaction between adsorbent and contaminant as a chemical adsorption process?

6. An explanation of the abbreviations should be provided

7. The drawings are not legible and introduce confusion, there is no uniform systematics of data marking throughout the article, too illustrative without introducing specific values. They do not give a scientific view. Some data in the drawings can be presented in the form of a table, for example fig. 2

8. The values of the data quoted in the text should be verified with their equivalents in the figures and tables, there are significant discrepancies. The presented trends and the lack of reliability in this respect should be corrected

9. In Fig. 7, you can analogously compare the energy parameters of the biochars prepared with commercial products

10. In Table 1, I propose to present the data for total and micropore

10. Please mark the explanation of Basic groups in table 2

11. Incomplete methodology, lack of determination of some parameters, e.g. ash, indications of the origin of commercial products

12. The conclusions drawn are too general

13. Generally, the study lacks interpretation of the obtained results and their comparison with the results of other researchers

Author Response

Reviewer 3

First of all, we would like to thank you for your kindly review and valuable suggestions that allowed us to improve significantly our manuscript.

 1.I propose to precisely define the title and the exact purpose of the work. The aim of the work is too general, while the title should emphasize biochars and the method of its acquisition

The title has been changed. The aim has been clarified.

  1. There are no conclusions in the abstract, for the reader it is not important in the foreground that the examined parameters changed in such a wide range.

The abstract has been improved according to Reviewer advice.

  1. In introduction there is no reference to methods used for biochar precursor preparation which may ultimately affect the activated carbon properties (surface area, porosity, and other)

A new References has been added and the Introduction has been modified.

  1. Please provide reference methods for the characterization of biochars and activated carbons. When setting parameters, how many repetitions were made.

Relevant information has been included in the manuscript

  1. Has the Temkin model been considered which considers the interaction between adsorbent and contaminant as a chemical adsorption process?

The Temkin model has been included and discussed in the revised manuscript.

  1. An explanation of the abbreviations should be provided.

All abbreviations have been explained.

  1. The drawings are not legible and introduce confusion, there is no uniform systematics of data marking throughout the article, too illustrative without introducing specific values. They do not give a scientific view. Some data in the drawings can be presented in the form of a table, for example fig. 2.

Figure 2 has been transformed into Table 1. The readability of drawings has been improved and the data overview has been expanded.

  1. The values of the data quoted in the text should be verified with their equivalents in the figures and tables, there are significant discrepancies. The presented trends and the lack of reliability in this respect should be corrected.

It has been checked and corrected.

  1. In Fig. 7, you can analogously compare the energy parameters of the biochars prepared with commercial products.

It has been done according to the Reviewer suggestion.

  1. In Table 1, I propose to present the data for total and micropore

It has been changed according to the Reviewer proposal.

  1. Please mark the explanation of Basic groups in table 2.

It has been explained.

  1. Incomplete methodology, lack of determination of some parameters, e.g. ash, indications of the origin of commercial products.

The relevant information has been added in the revised manuscript.

  1. The conclusions drawn are too general

Conclusions section has been improved.

  1. Generally, the study lacks interpretation of the obtained results and their comparison with the results of other researchers

It has been corrected. A comparison of our results with literature data has been done.

Round 2

Reviewer 2 Report

The authors made all the required modifications 

Reviewer 3 Report

I accept as amended, my comments have been taken into account.